# Real-Time Pharmacovigilance: Transforming Population-Based Monitoring of Post-Approval Vaccine Safety Through Rapid Cycle Analysis (RCA)—A Review of the Published Literature

**DOI:** 10.3390/ph18010080

**Published:** 2025-01-10

**Authors:** Sampada Gandhi, Michelle R. Iannacone, Andrea Leapley, Li Wang, Mwedusasa Mtenga, Muhammad Younus, Joanne Wu

**Affiliations:** Safety Surveillance Research, Worldwide Medical and Safety, Pfizer, Inc., New York, NY 10001-2192, USA; sampada.gandhi@pfizer.com (S.G.); michelle.iannacone@pfizer.com (M.R.I.); andrea.leapley@pfizer.com (A.L.); li.wang7@pfizer.com (L.W.); mwedusasa-bety.mtenga@pfizer.com (M.M.); muhammad.younus2@pfizer.com (M.Y.)

**Keywords:** rapid cycle analysis, sequential testing, near real-time monitoring, pharmacovigilance, post-approval drug safety, vaccine safety

## Abstract

**Background/Objectives**: Rapid cycle analysis (RCA) is an established and efficient methodology that has been traditionally utilized by United States health authorities to monitor post-approval vaccine safety. Initially developed in the Vaccine Safety Datalink (VSD) in early 2000s, RCA has evolved into a valuable approach for timely post-approval signal detection. Due to the availability of additional near real-time data sources and enhanced analytic approaches, the use of RCA has expanded. This narrative review provides an in-depth assessment of studies that utilized RCA for safety surveillance to detect and evaluate safety signals in post-approval vaccine monitoring. **Methods**: Embase and Medline were searched on 8 August 2024 to identify post-approval non-interventional vaccine safety studies using RCA or other near real-time surveillance methods published from 1 January 2018 to 31 July 2024. Data on study characteristics (e.g., study population, data source, outcomes) and RCA methodological characteristics (e.g., type of comparator, sequential testing method, confounding control method) were extracted from the eligible RCA studies. **Results**: Of 1128 articles screened, 18 RCA vaccine safety studies were included, of which 17 (94.4%) were conducted in the United States (US). Twelve (67%) aimed at signal detection and six (33%) conducted further signal evaluation. Over 60% examined COVID-19 vaccine safety, with half using VSD. Over 80% conducted the RCA weekly or monthly and about 78% of the studies used a database-specific historical comparator group. **Conclusions**: This review indicates that most of the published articles on the application of the RCA methodology in vaccine safety studies are based on research conducted in the US. With increasing availability of near real-time data sources and advanced analytic methods capabilities, RCA is expected to be more widely deployed as an active surveillance tool to complement traditional pharmacovigilance. Future studies should explore the extension of vaccine RCA methodology for non-vaccine medicinal products.

## 1. Introduction

Timely surveillance and decision-making are crucial in post-approval population-based vaccine safety monitoring, especially in the early post-marketing period. Originated in the VSD in early 2000s, RCA is an established and efficient methodology that has traditionally been utilized by the United States (US) health authorities to monitor post-approval vaccine safety. This methodology has enabled periodic, near real-time data collection, analysis, and review to promptly detect and characterize potential safety signals. Compared to the methods used in other pharmacoepidemiology study designs that require extended periods for data collection and signal identification, RCA facilitates quicker safety signal identification, allowing for faster communication of the vaccine’s post-approval benefit–risk profile.

The VSD is an active surveillance system established in 1990 as a collaboration between the Centers for Disease Control and Prevention (CDC) and several health maintenance organizations to monitor adverse events post-immunization [1]. The VSD’s approach to data ascertainment improved upon the limitations of passive surveillance systems (e.g., the Vaccine Adverse Event Reporting System (VAERS)), enabling it to be used to assess temporality and potential causality between vaccine exposure and subsequently occurring adverse events. The integration of RCA into the VSD in 2005 demonstrated early on its capability to report vaccine exposure associations in near real time with accuracy. In 2005, despite the increase in VAERS reports of Guillain–Barré syndrome (GBS) post-vaccination, the VSD’s RCA program reported no association between exposure to the, then newly licensed, quadrivalent meningococcal conjugate vaccine, Menactra, and GBS and further supported the continuation of vaccination activities [2]. Another early success in 2005 was that the VSD’s RCA program reported an increased risk of seizure associated with a combined measles, mumps, rubella (MMR), and varicella vaccine recommended for children 12 to 47 months old. This finding resulted in a change from co-administration to a separate administration of the MMR and varicella vaccines [2]. The VSD framework and the development of enhanced PV analytic methods, such as RCA, have continued to support the near real-time monitoring of adverse events post-vaccination and have proven to be a model for the use of these approaches in other data sets.

RCA relies primarily on sequential analysis to identify potential safety signals. The original method, the sequential probability ratio test (SPRT), was developed by Abraham Wald in 1945 [3], with the advantage that the test relied on a significantly smaller number of observations than other hypothesis tests that required a larger, predetermined number of observations, allowing researchers to decide more quickly whether to proceed with an experiment. However, the SPRT required a prespecified relative risk to establish the alternative hypothesis, and if chosen incorrectly, a delay in identifying a potential safety signal may occur or result in a type II error (i.e., the safety signal would be missed) [4]. Recognizing the test’s disadvantages for vaccine safety studies, the maximized sequential probability ratio test (MaxSPRT) was developed. The MaxSPRT and additional adaptations of the original SPRT have since been used to address the test’s limitations and are now utilized often in RCAs, particularly those assessing potential safety signals post-immunization.

RCA was integral during the COVID-19 pandemic for early signal detection of safety outcomes among patients receiving the newly developed COVID-19 vaccines [5,6]. With the increase in novel biologics to prevent or lessen the disease severity of viral infections, the utility of RCA beyond the setting of the VSD system is important for timely information on vaccine safety in the general population in the post-approval setting. Therefore, the aim of this narrative review was to provide a comprehensive and in-depth assessment of studies that utilized the RCA methodology to detect and characterize safety signals in post-approval vaccine monitoring and were published from January 2018 to July 2024. The findings from this review may offer valuable insights to pharmacoepidemiologists, drug safety experts, and PV professionals looking to leverage RCA for enhanced vaccine safety surveillance.

## 2. Results

Out of the 1128 articles that were identified and screened, 228 articles were eligible for a full-text review based on the review of titles and abstracts in Step 1 (Figure 1). Of the 228 articles, 37 were considered eligible after the full-text review in Step 2. The reasons for exclusion in each step are as outlined in Figure 1. Of the 37 eligible articles, 18 [5,6,7,8,9,10,11,12,13,14,15,16,17,18,19,20,21,22] vaccine safety RCA studies were included for in-depth review and data extraction, 13 [23,24,25,26,27,28,29,30,31,32,33,34,35] articles described approaches to define and apply RCA methodological characteristics (i.e., articles on RCA statistical methods), and 6 [36,37,38,39,40,41] articles summarized non-RCA near real-time surveillance methods.

### 2.1. Characteristics of Eligible Vaccine Safety RCA Studies (n = 18)

The characteristics of the 18 studies included in the in-depth review are summarized in Table 1. Most (n = 15, 83.3%) of the vaccine safety RCA studies were published in the post-COVID-19 pandemic era (i.e., in the years 2021–2024), whereas only three (16.7%) articles were published in the pre-pandemic era, in the year 2019. Most studies were conducted by either a regulatory agency (38.9%) or academic institution (33.3%). The product of interest was a vaccine in all studies, with 61.1% of the studies involving a COVID-19 vaccine. Of the total studies, 17 (89.5%) were conducted using a data source from the US, with the Vaccine Safety Datalink (VSD) being the most used data source (50%). The other data sources from the US that were used were Optum, CVS health, US Centers for Medicare and Medicaid Services, and Carelon Research/HealthCore (22.2% each). The most common type of data source was electronic health records, or EHRs (55.6%), followed by insurance claims (44.4%). Of the total studies, 44.4% studies examined <5 and 55.6% studies examined ≥5 adverse events of special interest (AESIs) related to vaccines of interest. The most common (n = 5, 27.8%) study population included in the studies was the pediatric age group of 0–17 years, followed by older adults (≥65 years), adults, and a combination of pediatric and adult populations (n = 3, 16.7%).

### 2.2. RCA Methodological Characteristics of the Eligible Vaccine Safety RCA Studies (n = 18)

Table 2 summarizes the key characteristics of the RCA methods used in the 18 studies included in this in-depth review. Of the total studies, 66.7% were conducted with the purpose of signal detection only and did not report further signal evaluation of the detected signals using a robust epidemiologic study design, whereas the remaining studies (33.3%) included a follow-up robust signal evaluation. For instance, Nelson et al. (2023) conducted an end-of-surveillance analysis to assess the robustness of the sequential results for the primary outcomes by using a complementary well-visit concurrent comparator group in a one-time set of analyses, adjusting for additional confounders using propensity scores [17]. The most employed frequency of sequential analyses was weekly (55.6%) followed by monthly (27.8%). Over 75% of the studies used data from a study period extending from <6 months to 2 years, reflecting the rapid nature of these analyses. The probability ratio test, in the form of maximized sequential probability ratio test (MaxSPRT), binomial-based version of MaxSPRT (BMaxSPRT), Poisson maximized sequential probability ratio test (PMaxSPRT), and Poisson-based conditional version of MaxSPRT (CMaxSPRT), was used to conduct sequential analyses in 61.1% of the studies, followed by conditional Poisson regression or Poisson regression (22.2%).

In 15 (83.3%) of the studies, a predefined signaling threshold was used to define a preliminary safety signal for an outcome, whereas in 3 (16.7%) of the studies, such a threshold was not specified by the authors. Two comparator groups were used in the studies as follows: (1) database-specific historical (background) rates (77.8%) and (2) concurrent (contemporaneous) comparators (44.4%). Within these two broad categories of comparator groups, variations of comparator groups were used, for example, historical rates estimated from the general population versus those estimated from vaccinated populations. In 33.3% of the studies, more than one type of comparator was used, including up to three comparator groups (e.g., Hanson et al., 2022) [9]. Covariate stratification was used as a method for control of confounding in 55.6% of the studies, followed by standardization of historical rates by age and sex (11.1%), conditioning on Poisson regression on strata (5.6%), and adjustment in modeling (5.6%). Overall, confounding adjustment in RCA was limited. Age, sex, race, ethnicity, site, calendar day, and type/brand/timing of vaccine administration were the most accounted for confounders in the analyses.

### 2.3. Characteristics of Selected Representative Vaccine Safety RCA Studies (n = 6)

Table 3 presents the study summaries and RCA characteristics of 6 representative studies selected from the pool of 18 eligible studies. These 6 studies were selected as they were considered representative with respect to either the study population (pregnant individuals, pediatric individuals, or older adults) included in the analysis [11,19,22] or the unique RCA methodological characteristics [9,14,17].

#### 2.3.1. The Following Studies Highlight the Various Study Populations

Pregnant individuals: Vazquez-Benitez et al. (2023) used a unique case–control surveillance design to rapidly detect and evaluate safety signals associated with COVID-19 vaccines in a pregnant population [22]. The study adapted a validated pregnancy algorithm to identify pregnancies in near real time for post-vaccination surveillance purposes in pregnant population.Pediatric population: Hu et al. (2024) conducted near real-time monitoring of health outcomes after COVID-19 vaccination in the pediatric population aged 6 months to 17 years, identified from three US commercial claims data sources [11]. The study included a large, geographically diverse population from these databases, which was essential for the examination of rare events such as myocarditis and pericarditis.Older adult population: Perez-Vilar et al. (2019) examined the risk of GBS following influenza vaccinations during the 2017–2018 season in US Medicare beneficiaries, primarily including adults 65 years and older [19]. The investigators conducted continuous weekly sequential testing to assess whether the observed GBS risk was elevated relative to the GBS rate from the five prior influenza seasons (historical comparator).

#### 2.3.2. The Following Studies Highlight Key RCA Methodological Characteristics

Historical comparator, multiple AESIs, multiple data sources: Lloyd et al. (2022) examined COVID-19 vaccine safety in over 16.8 million individuals aged 12–64 years identified from three large claims databases [14]. The study also provides a good example of the inclusion of multiple COVID-19 vaccine-related AESIs in the analysis. For comparison, the study used database-specific historical (background) rates from either a general population or an influenza-vaccinated population as a comparator group.Vaccinated and unvaccinated concurrent comparator, weekly frequency of analysis: Hanson et al. (2022) examined COVID-19 vaccine safety using vaccinated concurrent comparators defined as those similar vaccinated persons who were concurrently on the same calendar day in a post-vaccination comparison interval following the same vaccine type and unvaccinated concurrent comparators defined as those similar persons who were concurrently unvaccinated on the same calendar day [9].Confounding control using covariate stratification: Nelson et al. (2023) conducted signal detection using RCA and end-of-surveillance analyses for signal evaluation for recombinant zoster vaccine. The RCA component used historical and concurrent comparators as well as covariate stratification for confounding control [17]. Signal detection analysis employed covariate adjustment by estimating historical event rates stratified by site, age group, and sex. For two outcomes of interest, i.e., acute myocardial infarction and stroke, adjustment also included baseline diabetes and hypertension status. The end-of-surveillance signal evaluation analysis used a concurrent well-visit comparator and adjusted for confounding using propensity score matching.

The key study and RCA methods characteristics of the remainder of the 12 vaccine safety studies have been summarized in Appendix A (COVID-19 vaccine-related studies, n = 7) and Appendix A (non-COVID-19-related studies, n = 5).

### 2.4. Characteristics of Studies That Used Non-RCA Real-Time Surveillance Methods (n = 6)

Table 4 presents key study characteristics of six studies that used non-RCA real-time surveillance methods. All studies reported real-time surveillance methods used for examining vaccine safety (i.e., three COVID-19 vaccines, two any vaccines, and one influenza vaccine). The following real-time surveillance methods for data collection were reported in these studies: (1) primary data collection using a short message service (SMS) opt-out survey [37,39], (2) anonymized telephone helpline data [38], (3) online participant questionnaire completed up to six months following vaccination using two online platforms [40], (4) primary data collection via customized adverse drug reaction card [36], and (5) hierarchical diagnosis tree and tree-based scan statistics using the VSD database from the US [41]. The analytic methods used were descriptive in nature (counts and proportions) in at least three of the six reviewed studies, in contrast to the application of sequential testing methods used in the RCA studies [36,37,40].

Of the 13 studies [23,24,25,26,27,28,29,30,31,32,33,34,35] that provided an overview of the different aspects of statistical methods employed in the RCA (data not shown), at least 3 articles provided guidance on finding the optimal alpha spending and type I error spending [30,32,35]. A few authors summarized the expected time to signal and the use of a non-flat signaling threshold while using the conditional maximized sequential probability ratio test (CMaxSPRT) [33,34]. Shen et al., 2023, Cook et al., 2019, and Nelson et al., 2019, have explored new group sequential test procedures which account for seasonality and variation from historical controls or which implement regression/analysis-based confounder adjustment and/or weighting to control confounding [23,29,31].

## 3. Discussion

This review provides an in-depth overview of recently published studies that used the RCA methodology to detect safety signals following vaccine administration in near real-time in the post-approval setting. The findings of this review of the global literature indicate that most of the published articles on the RCA methodology are from studies conducted using claims and EHR data sources in the US, not only by regulatory agencies but also by academic institutions. The RCA approach has been used in study populations across all age groups (adults, pediatric 0–17 years, and older adults ≥ 65 years) and in pregnant individuals. However, no studies with an application of RCA to non-vaccine medicinal products were identified in this review. The authors considered the following two reasons for the more frequent use of the RCA for vaccine versus non-vaccine products: (1) RCA is better suited for vaccine products due to the transient nature of exposure and acute-onset outcomes (e.g., acute infections, anaphylaxis), which are anticipated to occur within a relatively well-defined risk window following vaccination compared to delayed-onset outcomes (e.g., renal impairment, malignancy) that require months or years following an exposure to a non-vaccine product, and (2) since vaccines are often recommended for use in healthy populations, a large number of exposures required to study safety signals, particularly rare events, accumulates quickly within a small time frame following vaccine approval. Other types of near real-time safety surveillance studies using various data collection methods, for example, primary data collection via SMS opt-out surveys and anonymized telephone helpline data, were also conducted.

Recently conducted vaccine RCA studies have demonstrated several noteworthy strengths. These studies used large, geographically diverse study populations identified from one or multiple real-world data sources across the US. The broad representation improves the generalizability of the research findings. Further, these studies utilized frequent data updates ensuring timely analyses and allowing for the rapid identification of potential safety concerns. Another key strength is the use of pre-specified signaling thresholds, which allows for the objective determination of whether a statistical signal is rejected or not. Moreover, RCA studies employed either contemporaneous or historical comparator groups or both, depending on the operational feasibility and rarity of safety outcomes of interest. One-third of the studies included in this review also conducted a follow-up signal evaluation after the initial RCA, using other epidemiologic methods with better control of confounding and provided a more robust assessment of vaccine safety.

The limitations of these RCA studies include a low statistical power to detect small risk increases due to insufficient sample size during the early post-approval period, particularly for rare outcomes. Further, limited control of confounding could lead to erroneous associations between the vaccine and the outcomes being studied. Also, an early warning system may falsely identify signals (false positives) due to multiple statistical tests performed, a possible mis-specification of RCA parameters, and early surveillance bias. This bias may potentially be influenced by high reporting and media attention surrounding potential safety concerns, as well as changes in the characteristics of the vaccinated population over time, since early vaccine recipients tend to be at a higher risk for developing safety outcomes. For example, for the Shingrix vaccine, the RCA signaled a 5-fold increased risk of GBS in the early vaccine rollout period, but the elevated risk did not sustain further into the surveillance [17]. Conversely, it is possible that true safety signals (false negatives) may be missed due to mis-specified parameters in the analyses (e.g., high signaling threshold) or rarity of the outcomes being examined. In addition, RCAs have limitations due to the use of near real-time administrative claims and EHR data, which may be subject to incomplete data regarding exposure or safety outcomes. Near real-time data do not guarantee complete data, and the potential for a data accrual lag, i.e., a time gap between the occurrence of safety outcomes and when these are recorded in the database, must be carefully considered. This lag can impact the validity of inferences regarding the rates of safety outcomes [17]. A common approach to account for data accrual lag is to delay the initiation of analysis by a specified period after the data cutoff date, allowing for at least 90% data completeness before proceeding with the analysis. For instance, a 12-week waiting period has been used in RCA studies in the VSD based on prior understanding of data lags in the data sources [17,18]. However, during the COVID-19 pandemic, when there was an urgent need for immediate safety results, researchers modeled data lag adjustment in the sequential analysis based on historical data lag patterns in the database to avoid any delays [13]. Lastly, for outcomes without validated case definitions, additional time may be needed for manual chart review to confirm outcomes of interest. Prior reviews suggested the potential of artificial intelligence and machine learning in expediting the timeliness of outcome review and improving the accuracy of outcome identification [42].

The choice of comparator groups for RCA requires careful consideration, as each type has its own strengths and limitations. Historical comparators are frequently used in evaluating rare outcomes, as was observed in 77.8% of the reviewed RCA studies. This is because stable, expected outcome rates in the comparator can be generated using multiple years of prior data, allowing for rapid assembly of the comparator group and enabling the analysis to begin immediately. However, a key limitation of using historical comparator rates is that the incidence proportions may fluctuate over time, posing challenges for observed versus expected sequential analysis. When historical comparator rates increase or decrease over time, the detected signal may be due to the background trends rather than the effect of the exposure of interest itself. Due to this known limitation of historical comparators, trends in background historical incidence proportions should be examined for each outcome of interest separately prior to undertaking RCA using historical comparator rates. If such trends are observed, the null hypothesis threshold for no excess risk could be adjusted to prevent false positive signals [43]. In some RCA studies included in this review, investigators examined whether rates differed substantially across historical periods [6,11,14,15]. These studies used the lowest or most stable historical rate to improve the sensitivity of sequential testing. On the other hand, concurrent comparator groups reduce concerns about background trends in the incidence of outcomes of interest, as individuals are selected contemporaneously with individuals from the exposed group. However, this type of comparator is generally slower to accrue and consequently slower to detect statistically significant findings, as it requires time to accrue outcomes in near real time in both the exposed and comparator groups. Notably, the vaccinated concurrent comparator (e.g., individuals who received the same vaccine but were in the post-vaccination control interval) emerged as a novel type of comparator in RCA studies on COVID-19 vaccines due to its improved efficiency, confounding control, and timeliness [13].

Although RCA is a valuable tool for enhanced vaccine safety surveillance, our literature search did not identify any published industry-led RCA analyses from 2018 to 2024. This may in part reflect the industry’s limited access to fit-for-purpose real-world data sources for RCA in the past. This review may offer valuable insights to investigators in the industry, encouraging the consideration of conducting RCA, particularly keeping in mind the growing availability of near real-time data sources and data networks. Some of the key elements for consideration in the feasibility assessment and design of RCAs include the following: (1) fit-for-purpose data source(s) for RCA should have frequent updates (e.g., weekly to quarterly); (2) the length of the surveillance period should be dictated by accruing an adequate number of exposures to achieve sufficient statistical power for detecting a prespecified risk(s) of high-priority safety outcomes; (3) RCAs are best suited for outcomes with a validated, codified case definition and with a known positive predictive value. Prompt chart review to validate outcomes may be needed for less well-defined outcomes; (4) concurrent comparator groups are preferred, where feasible, to avoid bias from time trends and calendar time. For rare events, historical comparators in the early phase could be utilized to attain statistical power; (5) the frequency of sequential testing in RCA is dependent on the rarity of safety events, frequency of data refreshes, and vaccine uptake post-approval. The signaling threshold should be pre-specified, which is usually achieved by an alpha-spending plan designed to keep overall chance of a Type I error (e.g., false positive) below 0.05 across the total number of sequential tests conducted; (6) investigators should consider including an end-of-surveillance epidemiologic study (e.g., cohort, SCRI) with better control for confounding, either using the same or an external data source, to further evaluate the findings from the RCA.

This review has several limitations. First, it included 18 studies, of which 17 were conducted using data sources within the US. As a result, researchers may need to consider whether the RCA methodology is applicable or feasible for studies using data sources outside the US. This review was not restricted only to studies conducted in the US. Also, given that this review only included published articles, studies conducted using non-US data sources that are not yet published were not captured in this review. Second, due to the timeframe of the literature search, there was an over-representation of COVID-19 vaccine RCA studies. Third, all included studies examined vaccine safety, which may limit the generalizability of these methodologies to non-vaccine medicinal products. Fourth, although the literature search was conducted comprehensively using two primary online databases, some RCA studies may have been missed due to the search terms used, which aimed to balance sensitivity and specificity. Lastly, as with all reviews, publication bias remains a concern, as RCA studies with negative findings may not be published.

## 4. Materials and Methods

### 4.1. Literature Search Strategy

On 8 August 2024, a comprehensive and systematic search of Embase and Medline was conducted via Ovid to identify all potential articles published from 1 January 2018 to 31 July 2024. This timeframe was deliberately chosen to include the COVID-19 pandemic era, which stimulated the use of RCAs to monitor the safety of COVID-19 vaccines, and the pre- and post-pandemic era, to allow for the representation of near real-time surveillance methods employed in non-COVID vaccine studies. The literature search strategy was developed in consultation with a medical librarian and included the following concepts: vaccines, safety surveillance/adverse event reporting, near real-time surveillance, and RCA. Text words for sensitive concepts like safety were limited to title and abstract while combining these logically with other related keywords. Similarly, text words for vaccines were limited to title and abstract to maintain the specificity of the subheadings attached to vaccine-related indexing. However, the concept of RCA was extended beyond title and abstract to capture potential headings and author keywords, as no indexing currently exists for this concept. The combination of these facets balanced the sensitivity and specificity of the literature search strategy. Non-English-language studies, animal model studies, and duplicate articles identified from both Medline and Embase were excluded. The specific search strings used for the literature search are included in Appendix A.

### 4.2. Screening Criteria and Selection of Studies

This review primarily focused on post-approval non-interventional vaccine safety studies that employed RCA methods. To be classified as a study utilizing the RCA method, the study was required to have 2 minimum pre-specified criteria: (1) evidence of frequent data collection or data refreshes during the surveillance period and (2) evidence of sequential analysis and/or testing of safety outcomes of interest to detect or assess a safety signal. Post-approval vaccine safety studies identified from the literature search that utilized non-RCA near real-time surveillance methods (as acknowledged by the authors) were also summarized for informational purposes, but those were not the focus of this review, as the search strategy was not specifically tailored to identify those studies.

After the initial search, all articles were downloaded to an Excel file and screened in 2 steps for the identification of eligible articles. During Step 1, study titles and abstracts were screened to exclude studies clearly outside the scope of this review, which included studies without vaccine or drug exposures, studies without the assessment of safety outcomes, and clinical trials/interventional studies. During Step 2, full-text articles were independently reviewed for inclusion based on the eligibility criteria by 2 experienced pharmacoepidemiologists. Discordance between the 2 pharmacoepidemiologists was resolved via a discussion and a 3rd pharmacoepidemiologist was further consulted for consensus if necessary. Articles excluded in Step 2 were cross-sectional surveys, case reports, case series, vaccine or drug safety studies without the application of RCA or any near real-time surveillance methods, review articles without relevant data, or irrelevant articles that were not excluded in Step 1 due to insufficient information from the study title and abstract. A reference list of prior vaccine review articles was also scrutinized to identify any relevant RCA studies that were published during the literature search period. Eligible studies were categorized into three groups: vaccine safety studies using RCA, articles describing approaches to define and apply RCA methodological characteristics (i.e., articles on RCA statistical methods), and vaccine safety studies that utilized non-RCA “near real-time surveillance methods”. Detailed data extraction was performed for vaccine safety studies using RCA only, as described in the section below. Article screening steps and reasons for exclusion are outlined in Figure 1.

### 4.3. Data Extraction, Analysis, and Presentation

The eligible vaccine safety studies were distributed among all epidemiologists. Data were extracted independently using a predefined structured template for vaccine RCA studies and included key information on study characteristics (e.g., authors, medicinal product, study population, geographical region, name and type of data source(s), safety outcomes of interest, sample size, and study period) and RCA methodological characteristics (e.g., types of comparator group(s), sequential testing methods, use of signaling detection threshold, confounding variables, methods used for control of confounding, and author-stated study strengths and limitations). The extracted data were verified by epidemiologists against the source data and no efforts were made to obtain additional data from the original study investigators. Data were summarized as counts and proportions and presented in a tabular format. The conduct of this systematic review followed the PRISMA guidelines.

## 5. Conclusions

Over the last decade, the approach to post-approval vaccine surveillance has experienced a substantial transformation, transitioning from traditional passive monitoring, which relied heavily on spontaneous reporting of voluntary reports from healthcare providers and patients to regulators and sponsors, to an increasingly active and systematic approach. While useful, the passive system is often limited by under-reporting, delays, and lack of detailed clinical information, potentially affecting the timely detection and evaluation of safety concerns. The shift towards an active surveillance framework, including RCAs and other near real-time surveillance methods, is largely driven by technological advancements and the availability of large-scale, rapidly accessible EHRs, insurance claims data, pharmacy records, and digital health platforms, which collectively provide a vast array of near real-time information. In the near future, advanced data integration and automated analytical capabilities, such as machine learning and artificial intelligence, are expected to further enhance signal detection and trend analysis that can identify safety signals with improved accuracy and speed. Overall, this proactive approach to vaccine surveillance offers substantial benefits. The ability to monitor vaccine safety in near real time allows for the early detection of potential safety signals, reducing the time lag between the occurrence of a safety outcome and its investigation. Moreover, active surveillance enables rapid response to emerging data, providing regulators and healthcare agencies with critical insights for ongoing risk–benefit assessments. Owing to the aforementioned advancements in big data availability, analytic approaches, and technology, RCA has the promising potential to grow as an active PV tool to complement traditional surveillance for vaccine safety, particularly in the industry setting. Future directions include the adaptation and extension of the RCA methodology to non-vaccine medicinal products.

## Figures and Tables

**Figure 1 pharmaceuticals-18-00080-f001:**
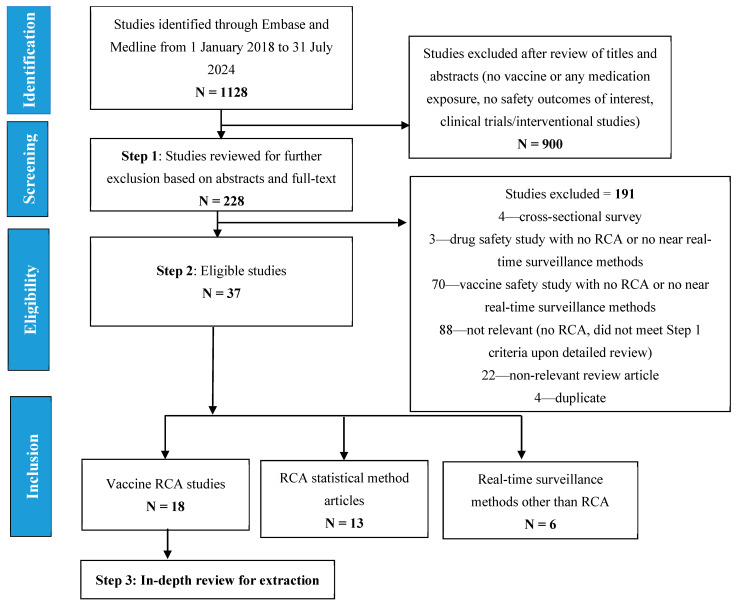
Flow chart showing the screening and selection of the studies included in this review.

**Table 1 pharmaceuticals-18-00080-t001:** Characteristics of eligible vaccine safety RCA studies identified in Step 2 (N = 18).

Characteristic	N (%)
Year of publication	
2018	0 (0)
2019	3 (16.7)
2020	0 (0)
2021	3 (16.7)
2022	4 (22.2)
2023	6 (33.3)
2024	2 (11.1)
Primary and Corresponding Author Affiliation	
Regulatory agency	7 (38.9)
Academic	6 (33.3)
Industry	0 (0)
Multiple	4 (22.2)
Other	1 (5.3)
Product	
Vaccine	18 (100)
COVID-19	11 (61.1)
Influenza	3 (16.7)
Other	4 (22.2)
Drugs	0 (0)
Geographical Location	
United States	17 (94.4)
Europe	0 (0)
Other	1 (5.6)
Data Sources *	
VSD	9 (50)
Optum	4 (22.2)
CVS Health	4 (22.2)
CMS	4 (22.2)
HealthCore/Carelon Research	4 (22.2)
VHA	1 (5.6)
Other	3 (16.7)
Type of Data Source	
Insurance claims	8 (44.4)
Electronic health records	10 (55.6)
Outcomes	
Specific AESIs (<5)	8 (44.4)
Multiple AESIs (≥5)	10 (55.6)
Study Population	
Any age	2 (11.1)
Pediatric (0–17 years)	5 (27.8)
Older adults (≥65 years)	3 (16.7)
Adults (18–64 years)	3 (16.7)
≥12 years	2 (11.1)
Pediatric and adult	3 (16.7)
Description of Population	
General healthy population	17 (94.4)
Pregnant	1 (5.6)
Immunocompromised	0 (0)

Abbreviations: VSD, Vaccine Safety Datalink; VHA, Veterans Health Administration; CMS, US Centers for Medicare and Medicaid Services; COVID-19, COronaVIrus Disease of 2019; AESIs, adverse events of special interest. * Some studies used multiple data sources. Therefore, the numbers do not add up to the total 18 extracted vaccine RCA studies.

**Table 2 pharmaceuticals-18-00080-t002:** RCA methodological characteristics of the eligible vaccine safety RCA studies (N = 18).

Characteristic of RCA	N (%)
Purpose of RCA	
Signal detection only	12 (66.7)
Signal evaluation only	0 (0%)
Signal detection and evaluation	6 (33.3)
Frequency of RCA	
Weekly	10 (55.6)
Monthly	5 (27.8)
Biweekly or monthly	1 (5.6)
Not specified	2 (11.1)
Duration of Study Period	
<6 months	1 (5.6)
6 months–1 year	8 (44.4)
1–2 years	5 (27.8)
>2 years	3 (16.7)
Not specified	1 (5.6)
Statistical Analysis Method	
One-sided Fisher’s exact test and exact logistic regression analysis	1 (5.6)
Conditional Poisson or Poisson regression	4 (22.2)
Adjusted odds ratios	1 (5.6)
Exact sequential Poisson-based likelihood ratio test	1 (5.6)
Probability ratio test (MaxSPRT/BMaxSPRT/PMaxSPRT/CMaxSPRT)	11 (61.1)
Pre-specified Signaling Threshold	
Yes	15 (83.3)
Not specified	3 (16.7)
Type of Comparator *	
Database-specific historical (background) rates	14 (77.8)
Historical rates estimated from all population	10 (55.6)
Historical rates estimated from vaccinated population	6 (33.3)
Concurrent comparators	8 (44.4)
Vaccinated (same vaccine as exposed) comparator	4 (22.2)
Vaccinated (other vaccines) comparator	1 (5.6)
Unvaccinated comparator	3 (16.7)
Self-controlled (e.g., SCRI)	2 (11.1)
Case–control	1 (5.6)
≥1 comparator	6 (33.3)
Method for Control of Confounding *	
Covariate stratification	10 (55.6)
Conditioning on Poisson regression on strata	1 (5.6)
Standardization of historical rates	2 (11.1)
Adjustment in modeling	1 (5.6)
Not specified	5 (27.8)
Confounding Variables *	
Age	16 (88.9)
Sex	14 (77.8)
Race/ethnicity	8 (44.4)
Season	1 (5.6)
Site	7 (38.9)
Region	2 (11.1)
Calendar day	4 (22.2)
Type/brand/timing of administration of vaccine	4 (22.2)
Claims processing delay	3 (16.7)
Concomitant vaccination	2 (11.1)
Comorbidities	1 (5.6)
Nursing home residency status	2 (11.1)
Other	2 (11.1)
Not specified/None	2 (11.1)

Abbreviations: MaxSPRT, maximized sequential probability ratio test; BMaxSPRT, binomial-based version of MaxSPRT; PMaxSPRT, Poisson maximized sequential probability ratio test; CMaxSPRT, Poisson-based conditional version of MaxSPRT. * The total does not round up to the total number of 18 extracted vaccine RCA studies, as some studies used more than 1 comparator, used more than 1 method for control of confounding, or controlled for more than 1 confounder.

**Table 3 pharmaceuticals-18-00080-t003:** Study and RCA characteristics of 6 representative vaccine safety RCA studies.

Author, Year and Country	Name of Data Source (Type)	Vaccine Studied	Purpose of RCA	Study Period	Study Population	Safety Outcomes *	Type of Comparator	Statistical Analysis Method	Frequency of Analysis	Signaling Detection Threshold	Confounding Control Method	Confounding Variables
Criteria for study selection: Pediatric population												
Hu, 2024, USA [11]	Optum, Carelon Research, and CVS Health supplemented vaccination data from local and state Immunization Information System (IIS) (Claims)	COVID-19	Signal detection	December 2020 to April 2023	Pediatric general population	* Anaphylaxis; appendicitis; Bell’s palsy; common site thrombosis with thrombocytopenia; deep vein thrombosis	Historical comparator cohort; used database-specific historical (expected) rates	Poisson maximized sequential probability ratio test (PMaxSPRT)	Monthly	Yes	Stratification of outcome rates in the vaccinated population; historical rates adjusted for claims processing delay and standardized by age and sex	Stratification by age, sex, region, urban/rural status, data source, and vaccine brand
Criteria for study selection: Pregnant population, use of case–control for RCA, signal detection												
Vazquez Benitez, 2023, USA [22]	Vaccine Safety Datalink (VSD) (electronic health records)	COVID-19	Signal detection and evaluation	December 2020 to July 2021	Adult pregnant individuals	Spontaneous abortion (SAB)	Case–control study design; controls were ongoing pregnancies of less than 20 weeks’ gestation	Adjusted odds ratios for case–control surveillance design	Monthly	Not specified	Adjustment in modeling	Maternal age at pregnancy start date, race/ethnicity, gestational age at SAB or index date, number of prenatal care visits up to SAB or index date, VSD site, surveillance period
Criteria for study selection: Older adult study population, use of Medicare data, non-COVID-19 vaccine												
Perez Vilar, 2019, USA [19]	Centers for Medicare and Medicaid Services	Influenza	Signal detection and evaluation	August 2017 to June 2018	Older adult general population	Guillain–Barré syndrome (GBS)	Historical rates from five prior seasons in the RCA and self-controlled risk interval (SCRI) for signal evaluation	Updating sequential probability ratio test (USPRT) for RCA, conditional logistic regression for SCRI	Weekly	Yes	Subgroup analysis in RCA; stratified analysis in SCRI	Age, type of flu vaccine in RCA; age, sex, type of flu vaccine, concomitant vaccination in SCRI
Criteria for study selection: Vaccinated and unvaccinated concurrent comparator, weekly frequency of analysis												
Hanson, 2022, USA [9]	VSD (electronic health records)	COVID-19	Signal detection	December 2020 to November 2021	≥12 years general population	GBS	1. Similar vaccinated persons who were concurrently in a post-vaccination comparison interval; 2. Similar persons who were concurrently unvaccinated; 3. Historical background rate of GBS	Conditional Poisson regression one-sided sequential testing	Weekly	Yes	Conditioning the Poisson regression on strata	5-year age group, sex, race, and ethnicity, site, calendar day
Criteria for study selection: COVID vaccine, large sample size, historical comparator, multiple AESIs, multiple data sources												
Lloyd, 2022, USA [14]	Optum, HealthCore, CVS Health (claims supplemented with immunization information system vaccination data)	COVID-19	Signal detection	December 2020 to January 2022	12–64 years general population	* Acute myocardial infarction, deep vein thrombosis, pulmonary embolism, disseminated intravascular coagulation, non-hemorrhagic stroke	Historical comparator cohort (either general population or influenza- vaccinated) from pre-pandemic period (Optum: 2019; CVS health: 2019 or June–December 2020; HealthCore: 2017, 2018 or 2019)	Poisson maximized sequential probability ratio test	Biweekly or monthly	Yes	Covariate stratification by age and sex	Age, sex
Criteria for study selection: Confounding control using covariate stratification												
Nelson, 2023, USA [17]	VSD (Electronic health records)	Recombinant zoster vaccine (RZV)	Signal detection and evaluation	January 2018 to December 2019	Adult ≥50 years general population	* Primary: acute myocardial infarction, stroke, supraventricular tachycardia, polymyalgia rheumatica, convulsions, Bell’s palsy	1. Historical zoster vaccine live (ZVL) comparator, 2. concurrent well-visit comparator	Exact sequential Poisson-based likelihood ratio test	Monthly	Yes	Covariate stratification	Site, age group, and sex. For acute myocardial infarction and stroke, adjustment included baseline hypertension and diabetes status. End-of-surveillance analyses adjusted for confounding using propensity score matching of RZV with well-visit concurrent comparators

* Only up to 5 safety outcomes are listed in the table. For studies with more than 5 outcomes as indicated by an asterisk, please refer to the original article for a comprehensive list of outcomes.

**Table 4 pharmaceuticals-18-00080-t004:** Key study characteristics of studies using non-RCA near real-time surveillance methods (N = 6).

Author, Year and Country(ies)	Data Sources and Period	Study Design	Study Population	Vaccine or Drug Studied	Safety Outcomes *	Analysis Approach and Signal Detection Method	Confounding Control Method and Confounding Variables (as Applicable)
Mesfin 2020, Australia [38]	Anonymized telephone helpline data, 2009–2017	Retrospective study using deidentified data	Residents of Victoria, Australia	Any vaccine	Weekly number of adverse event following immunization (AEFI)-related calls made to the nurse-on-call system	Using the Farrington surveillance algorithm, a Poisson generalized linear model calculated the expected count of AEFI calls for the current week based on historical data. A signal was declared if the examined week’s observed AEFI calls exceeded the upper bound (99% level) of the expected AEFI calls (threshold) for that week.	Seasonality
Raethke 2024, multiple European countries [40]	Two online platforms These were adapted for use in multiple EU countries February 2021 to February 2023	Multiple European countries: the Netherlands, Belgium, France, the United Kingdom, Italy, Portugal, Romania, Slovakia, and Spain	General population	COVID-19 vaccine	* Adverse reactions that were known to occur frequently after vaccination, especially fever/feverishness, shivering/chills, headache, nausea, myalgia/muscle pain,	Participants registered online within two days of receiving the vaccine and reported demographic data, medical history in a baseline questionnaire, and adverse drug reactions up to six months after vaccination in six follow-up questionnaires. Descriptive statistics reported proportions with 95% CIs.	None
Phillips 2021, Australia [39]	AusVaxSafety, collection of survey data from individuals following routinely administered vaccines at sentinel sites across Australia, 11/2016–11/2018	Opt-out survey sent via automated short message service (SMS) or opt-in online survey; cohort study	Adults 70–79 years	Zoster vaccine live	* fever, swelling or redness at injection site, pain at injection site, tired/fatigue, irritable	The study utilized fast initial response cumulative summation (FIRCUSUM) and Bayesian updating analyses. Phone survey on AEFIs was administered 3 days after vaccination. Results were reported fortnightly.	None
Yih 2023, USA [41]	Vaccine Safety Datalink (VSD), 1 August 2022–30 November 2022	Near real-time retrospective cohort	all ages combined and for the following age groups: <18, 18–39, 40–64, and >/= 65 years	Moderna CVX Code 229 and Pfizer-BionTech CVX Code 300, 301	>60,000 possible adverse events after bivalent COVID vaccination	The study utilized hierarchical diagnosis tree and tree-based scan statistics, specifically conditional self-controlled tree-temporal scan statistics, conditioning the expected distribution on both the total # of cases of each diagnosis or group of related diagnoses during follow-up and the total number of cases of any diagnosis occurring during the scanning window.	Self-control
Guedel 2021, Switzerland [37]	Primary data collection via cross-sectional opt-out short message service (SMS) survey, February–September 2020	Survey (using smart phone app SmartVax software)	Adults	Routine vaccines	Adverse events following immunization (AEFIs)	The study provided descriptive statistics of response rate, time-to-response, and frequency and type of AEFIs by vaccine and clinical subgroup	None
Dos Santos 2022, Spain, Germany, Belgium [36]	Primary data collection via customized adverse drug reaction (ADR) card to collect post-vaccination AEs, October 2020 to January 2021	Primary data collection	Any age, general population	GSK’s Quadrivalent Seasonal Influenza vaccine	* Any, general disorders and administration site conditions, injection site pain, injection site swelling, fatigue	The study provided descriptive statistics (cumulative number of participants by week, dose, and country, with the proportion reporting AE(s) within 7 days of vaccination)	Stratified analysis by select variables including age and country

* Only up to 5 safety outcomes are listed in the table. For studies with more than 5 outcomes as indicated by an asterisk, please refer to the original article for a comprehensive list of outcomes.

## Data Availability

No new data were created or analyzed in this study. Data sharing is not applicable to this article.

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
