# Peer review of "Real-Time Pharmacovigilance: Transforming Population-Based Monitoring of Post-Approval Vaccine Safety Through Rapid Cycle Analysis (RCA)—A Review of the Published Literature"

_pharmaceuticals, 2025, doi:10.3390/ph18010080_

Round 1
Reviewer 1 Report
Comments and Suggestions for Authors
Thank you for giving me the opportunity to comment on this submission. I understand that this review has focused mainly on US based vaccine studies.
I would appreciate some clarification in the Introduction or Discussion please, regarding these aspects:
1. Why has this methodology not been deployed internationally? What specific data capturing aspects are unique to the US which are not available elsewhere?
2. Similarly, why is there a paucity of RCA studies that are non-vaccine based? What are the specific issues that make vaccines more easily studied with this method, whereas other multiple dose or long-term treatments are not adequately covered?
3. Following on from the above, I see that there are several recommendations in the Discussion section regarding optimising the methodology, I would like to see some mention of the cost and resource requirements for running an RCA programme, especially if you feel that routinely collected codified claims data is less than optimal.
Also, Is the US-based vaccine focus due to the search limitations in this review? This is a major weakness leading to lack of generalizability of this review.
Minor points - lines 91-93 should be deleted.
Reviewer 2 Report
Comments and Suggestions for Authors
I read with interest the paper titled "Real-Time Pharmacovigilance: Transforming Population-Based Monitoring of Post Approval Vaccine Safety Through Rapid Cycle Analysis (RCA)- Requirements, Current Approaches, Advantages and Limitations"
The paper provide a good analysis of real time pharmacovigilance to vaccines. I have some comments that authors could use to improve the manuscript:
1. Please clarify the aim of the study in the end of the introduction.
2. Results first 3 lines (91-93) should be removed.
3. Chapter 2.3.1 and 2.3.2 could be transformed in tables.
4. Table 3 and Table 4 could be improved in terms of presentation. Too long.
5. Discussion is quite long and very intensive to read. Please provide fewer text and focus on the discussion of the findings of the paper.
6. Please provide PROSPERO code for systematic review.
7. Please follow PRISMA statement for the systematic review presentation.
8. Please present Risk of Bias evaluation for systematic reviews, as per PRISMA statement.
9. Conclusion is too long. Reduce it to focus on your objectives.
Round 2
Reviewer 2 Report
Comments and Suggestions for Authors
1 - PRISMA checklist should be provided stating in each page every step was fulfilled.
2 - Risk of bias is not an optional thing for systematic reviews and if authors followed the PRISMA properly, this should be included in the paper. The quality assessment aims to assess the "risks to rigor" present in a primary qualitative study by examining a study's methodological strengths and limitations, including research conduct. It is not linked with quantitative data/meta-analysis. This is related with the articles that you choose and that must be assessed by quality. If the authors choose not to fulfill the entire process of a systematic review, this cannot be called a systematic review, maybe a narrative one with a more "structured methodology".
Round 3
Reviewer 2 Report
Comments and Suggestions for Authors
No further comments to add. Accept in the current form.